# Analysis of the Spatial Correlation Network Structure of Agricultural Water Use Efficiency in Northwest China

**Yun Gao, Yulong Zhao \*, Kaiyang Li, Xuebin Qi and Ping Li \***

Farmland Irrigation Research Institute, Chinese Academy of Agricultural Sciences, Xinxiang 453002, China; gaoyun@caas.cn (Y.G.); likaiyang01@caas.cn (K.L.); qixuebin@caas.cn (X.Q.)

\* Correspondence: zhaoyulong@caas.cn (Y.Z.); liping05@caas.cn (P.L.)

**Abstract:** There are some problems in Northwest China, such as the fragile ecological environment, poor basic conditions of agricultural production, low efficiency of agricultural water use, and difficulty in clarifying the path of agricultural water use efficiency (AWUE) improvement. Based on the superefficiency data envelopment analysis (DEA) model, this study increases the amount of 'green water' resources in the agricultural water consumption index of the input index and increases the wastewater pollutants (total chemical oxygen demand emissions; total ammonia nitrogen emissions) in the undesired output index to measure the AWUE in the northwest region. Based on the calculation results of AWUE, combined with the modified gravity model, the connection strength of AWUE between any two provinces in Northwest China is calculated, and the spatial structure and network characteristics of AWUE in this area are analysed via the social network analysis (SNA) method. The results show that the average AWUE in 2020 is nearly two times higher than that in 2011. From the situation of the northwest provinces, the average AWUE of the five provinces is in the order of Qinghai > Shaanxi > Gansu > Ningxia > Xinjiang. The AWUE value, total population, real GDP and per capita GDP of the capital cities of the five provinces in Northwest China are the key factors for the improvement of the connection intensity of AWUE. From 2011 to 2020, the connection intensity and closeness of AWUE in Northwest China increased, and there was a clear network hierarchy. The improvement in overall AWUE in the region is mainly due to the radiation and driving effect of the central province on other provinces. Based on this, the study proposes policy recommendations for gradually realising the AWUE improvement path of the central province (Shaanxi; Gansu)—cooperation circle (Shaanxi–Ningxia; Gansu–Qinghai)—the whole region. The results provide theoretical support and a quantitative basis for optimising the spatial pattern of agricultural water resources and improving AWUE in Northwest China.

**Keywords:** Northwest China; AWUE; spatial association network structure; 'green water' resources; indicators of undesirable output

## 1. Introduction

Precipitation is the main factor affecting agricultural water consumption. The abundance of water directly affects the yield and quality of crops and affects the evaluation results of AWUE. According to the concept of ecohydrology, 'green water' is derived from precipitation stored in the soil of plant roots. It is stored in the soil and enters the atmosphere through evaporation and transpiration to form water vapour. Studies have shown that 80% of global food production is attributed to the effective precipitation of 'green water' [1,2]. The concept of 'green water' is relative to that of 'blue water' in rivers, lakes and underground aquifers. Most of the existing studies on AWUE only consider current statistical agricultural water consumption (that is, the amount of water used for agriculture by 'blue water' resources), ignoring the role of 'green water' resources in agricultural soil, making the consideration of AWUE not comprehensive enough. In fact, 'green water' is a major component of water resources. Introducing 'green water' into the category of

agricultural water resources is not only an inevitable requirement for food security but also necessary for a comprehensive and true understanding of AWUE [3,4].

Regarding the measurement method of AWUE, water consumption per ten thousand yuan GDP is a common index, which is usually measured via the ratio of the output to corresponding water consumption [3]. However, when using the unit output to evaluate AWUE, there is an implicit assumption that the output is created only by water resources designated as the input factor, which will lead to bias in the evaluation of AWUE because this method ignores the contribution of other production factors. Based on the inclusion of multifactor input into AWUE by Hu et al. [5], the measurement methods commonly used at present include principal component analysis [6], the analytic hierarchy process [7], the projection pursuit method [8], stochastic frontier analysis [9] and data envelope analysis [10], among which data envelope analysis is widely used as a nonparametric method. Because traditional measurement methods cannot overcome shortcomings such as the complex cross-relationship between various factors and the subjective weight assumption in the operation process [11], Charnes [12], Tone [13] and Zhou [14] et al. improved the model and proposed a superefficient data envelopment analysis (DEA) model with an unexpected output. The relaxation of input and output variables was achieved in this model, and the bias in efficiency measurement was minimised. The superefficient DEA model is one of several methods based on data envelopment analysis, which can measure the efficiency of all decision units and the relaxation of input and output variables. Using a superefficiency DEA model, Wang et al. [15] and Yang et al. [16] introduced unexpected parameters into environmental efficiency and environmental total factor productivity and conducted an empirical study on their influencing factors. Ma et al. [17] used an input-oriented DEA model to calculate the total factor AWUE containing undesired output and used a Tobit regression model to analyse the factors influencing AWUE in China and its different regions. Sun et al. [18] measured the water resource use efficiency of 31 provinces, municipalities and autonomous regions in China from 1997 to 2011 by using data envelopment analysis with and without a consideration of the unexpected output based on panel data on the interprovincial water footprint and grey water footprint.

Studies on the spatial–temporal differentiation of AWUE mostly focus on spatial correlation. Most scholars use exploratory spatial data analysis (ESDA) to study the spatial relationship related to geographic spatial location [19–22], but ESDA and spatial measurement techniques are limited by the measurement of proximity or distance relationships between regions in geographic space. It is difficult to dynamically grasp the structural characteristics of the spatial association of interprovincial resource utilisation as a whole. Moreover, as the interregional spatial flow and intercommunication of resource elements increase, the spatial association of resource utilisation presents the structural characteristics of multithreading and a complex network, which makes the spatial association form a 'relational data' matrix network between two pairs. As a result, the existing traditional econometric model based on 'attribute data' causes difficulty in fully revealing the overall network structure and spatial clustering mode of relational data. The social network analysis (SNA) method can break through the limitations of attribute data analysis and carry out effective analysis of the network characteristics of relational data. This method is an effective means with which to study the structural characteristics of relational data networks and has gradually become a new research paradigm in economics, management and other fields [23]. In recent years, scholars at home and abroad have revealed how individual social relations aggregate into complex structures through SNA methods, creating opportunities or limitations for the sharing and distribution of food resources [24]. The SNA method was used to study the social structure of animal groups and populations [25] and to study the social network composition of the rural elderly in South Africa [26], and based on the conflict detection and elimination decision process of SNA, a large-scale group decision model was proposed [27]. In the field of water resources, Deng et al. [28] used the SNA method to study the virtual water trading network of 19 countries and found that the virtual water trading network formed by the first sector is more intensive.

The northwest region of China is gradually becoming the forefront of opening up. The development of this region is already in a transitional stage of spatial order reconstruction and has a direct impact on China's economic development. However, the land resources in Northwest China account for nearly 1/3 of those in the country, and the total water resources in Northwest China are only 202.7 billion $m^3$, accounting for 7.25% of the country's total water resources [29]. The misallocation of water and land resources seriously restricts economic and social development. The northwest region has a shortage of resources, the agricultural water use efficiency (AWUE) is lower than the national average level, and the difference in AWUE among the five provinces (Qinghai, Shaanxi, Gansu, Ningxia and Xinjiang provinces) is extremely significant. The comprehensive irrigation quota is 7575 $m^3/hm^2$, which is 1.29 times the national average. The GDP output with 1 $m^3$ of water is 55 yuan/$m^3$, which is only 49% of the national average output and only 45 yuan/$m^3$ in Ningxia [30]. In summary, the uneven spatial and temporal distribution of water resources and the low AWUE have become serious problems in the northwest region.

Based on panel data from 2011 to 2020, this study evaluated the AWUE in Northwest China under the condition of climate change factors such as precipitation, with a super-efficiency DEA model that considered 'green water' and undesirable output index. The social network analysis method was used to further discuss the spatial network structure characteristics of AWUE in Northwest China to provide suggestions for optimising the spatial pattern of water resources and improving AWUE.

## 2. Research Methods and Data

### 2.1. Overview of the Study Area

In accordance with the natural geographical characteristics of Northwest China (Kun-lun Mountain, Altun Mountain (Qilian Mountain) and the arid and semiarid area north of the Qinling Mountains), this paper takes five provinces (regions) as the study area: Xinjiang, Gansu, Qinghai, Ningxia and Shaanxi (Figure 1). This region is located in the hinterland of China, with sufficient light and heat resources, rich land resources and low population density; however, in the study area, water resources are experiencing a severe shortage with uneven spatial and temporal distributions, the quality of cultivated land is low, and the ecological environment is fragile.

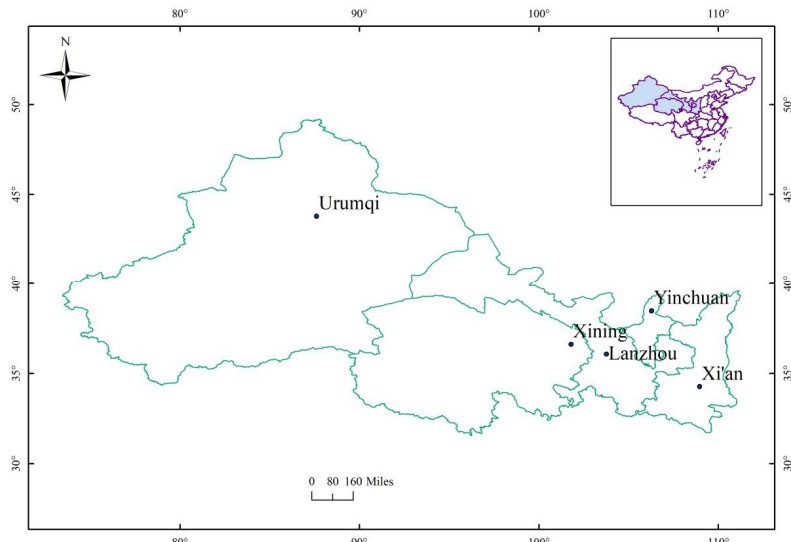

**Figure 1.** Location distribution in Northwest China. Legend: the green solid line in the figure is the provincial boundary; Xi'an, Lanzhou, Xining, Yinchuan and Urumqi are the capital cities of Shaanxi, Gansu, Qinghai, Ningxia and Xinjiang, respectively, and are the calculation base of the economic and geographical distance of the five provinces. The blue highlighted area in the data frame is the geographical location map of five provinces in Northwest China. The purple solid line is the boundary line of each province.

*2.2. Research Methods*

2.2.1. Superefficiency DEA Model with the Green Water Resource Input Variable Added

The superefficiency DEA model is a nonradial and nonangular DEA analysis method based on the measurement of slack variables. Based on the superefficiency DEA model, the indicators of green water resources and undesirable output (namely, the discharge of wastewater pollutants) are introduced into the evaluation system of AWUE in Northwest China. The input variables of the model include agricultural water consumption (including the amount of 'green water' resources), labour, sown area, agricultural machinery and fertiliser consumption. The desired output indexes include the total agricultural output value and the undesirable output index: total wastewater discharge (including total chemical oxygen demand discharge, total ammonia nitrogen discharge and other wastewater pollutant discharge). The number of 'green water' resources was calculated based on the actual precipitation in Northwest China from 2011 to 2020 by using the currently recognised and generally recommended method of the Soil Conservation Service of the United States Department of Agriculture [31].

The objective function of the superefficiency DEA model is as follows:

$$\min = \frac{\frac{1}{m}\sum_{i=1}^{m}(\overline{x}/x_{ik})}{1/(r_1+r_2) \times (\sum_{p=1}^{r1}\overline{y^d}/y_{pk}^d + \sum_{q=1}^{r2}\overline{y^u}/y_{qk}^u)}$$

where the constraint condition is as follows:

$$\overline{x} \geq \sum_{j=1,\neq k}^{n} x_{ij}\lambda_j, \quad i = 1, \ldots, m;$$

$$\overline{y^d} \leq \sum_{j=1,\neq k}^{n} y_{pj}^d\lambda_j, \quad p = 1,\ldots,r_1;$$

$$\overline{y^u} \geq \sum_{j=1,\neq k}^{n} y_{qj}^u\lambda_j, \quad q = 1,\ldots,r_2;$$

$$\lambda_j \geq 0, \quad j = 1, \ldots, n, j \neq 0;$$

$$\overline{x} \geq x_k, \quad k = 1,\ldots,m;$$

$$\overline{y^d} \leq y_k^d, \quad d = 1,\ldots,r_1;$$

$$\overline{y^u} \leq y_k^u, \quad u = 1,\ldots,r_2.$$

where *n* is the decision-making unit, and each decision-making unit has *m* inputs, the desired output, $r_1$, and the undesired output, $r_2$. X (X $\in$ R$^m$), Y$^d$ (Y$^d$ $\in$ R$^{r1}$) and Y$^u$ (Y$^u$ $\in$ R$^{r2}$) are treated as the matrix, X = [$x_1\ldots, x_n$] $\in$ R$^{m \times n}$, Y$^d$ = [$y^d_1\ldots, y^d_n$] $\in$ R$^{r1 \times n}$ and Y$^u$ = [$y^u_1\ldots, y^u_n$] $\in$ R$^{r2 \times n}$. It is generally believed that an efficiency value of 1 means that DEA is effective, an efficiency value of 0.8~1 (excluding 1) means that weak DEA is effective, and a value of less than 0.8 means that it is invalid [32].

Calculation of "green water" resources:

$$P_e = \begin{cases} \frac{1}{1500} \times P \times (1500 - 0.2 \times P) & P < 3000 \\ 1500 + 0.1 \times P & P \geq 3000 \end{cases}$$

where $P_e$ is the effective precipitation in each year from 2011 to 2020, in mm, and $P$ is the actual precipitation in each year, in mm.

2.2.2. Modified Gravity Model

The most commonly used methods to determine the spatial correlation are the gravity model and VAR Granger causality test. After a comprehensive consideration of the economic and geographical distance factors of the capital cities of five provinces in Northwest China, this study uses a modified gravity model to determine the spatial correlation between five provinces in Northwest China, which can reveal the dynamic evolution trend of the spatial correlation network of AWUE. The specific model equation is as follows:

$$y_{ij} = k_{ij} \frac{\sqrt[3]{P_i E_i G_i} \sqrt[3]{P_j E_j G_j}}{\left(\frac{D_{ij}}{g_i - g_j}\right)^2}$$

where

$$k_{ij} = \frac{E_i}{E_i + E_j}$$

where $i$ and $j$ represent five provinces in Northwest China, $y_{ij}$ is the attractiveness between the AWUE of the capital cities of five provinces $i$ and $j$, $E_{ij}$ represents the AWUE values of city $i$ and city $j$, $P_i$ and $P_j$ are the total population of city $i$ and city $j$ at the end of the year, $G_i$ and $G_j$ are the real GDP of city $i$ and city $j$, $k_{ij}$ is the contribution rate of city $i$ in the link of AWUE between city $i$ and city $j$, $D_{ij}$ represents the distance between any two cities, and $g_i - g_j$ represents the difference between the per capita GDP of the two cities.

2.2.3. Social Network Analysis Method (SNA)

SNA integrates mathematical operations and graph theory tools to quantitatively analyse the network of relationships among actors. The unit of analysis is both the structure of the relationship between the nodes of the network and the model of the relationship between the actors. Specifically, it studies how these structures have an impact on the behaviour and extent of network members. According to the research purpose, the representation of the social network structure can be divided into overall network feature analysis and individual network feature analysis, which can not only reflect the characteristics of the whole network structure but also reveal the status of individuals in the network structure. The study analyses the overall network characteristics of the spatial correlation of AWUE in the region through parameter indicators such as network density, core–periphery structure, and cohesive subgroups. The degree centrality (Appendix A), network connection direction, and betweenness centrality (Appendix B) indicators are used to describe whether or not the nodes are in the centre position in the network. Among them, network density refers to the closeness of the connection between nodes in a network graph, and the value range is [0, 1]. The greater the network density, the greater the influence on each node, and the higher the closeness of the connection. The cohesive subgroup can explain the substructure within a group and is a concept of the actor subset with a wide range of meanings. There are relatively strong, direct, close, frequent or positive relationships among actors in this set. This indicator can reveal which provinces in Northwest China have closer connections and the impact of this agglomeration on the linkage network of AWUE in the whole region.

*2.3. Construction of the AWUE Evaluation Index System and Data Source*

Based on the superefficiency DEA model, the 'green water' resource and the undesirable output, namely, the indicators of wastewater pollutants (total chemical oxygen demand discharge and total ammonia nitrogen discharge), are incorporated into the evaluation system of AWUE in Northwest China (Table 1). The basic data used in the calculation of the linkage strength of AWUE are regional GDP, per capita GDP, and total population, which are from the China Social Statistical Yearbook (2011–2020) [33].

**Table 1.** The evaluation system of WUE in Northwest China.

| Types of Indicators | Selection of Indicators | Data Source |
|---|---|---|
| Input indicators | Agricultural water consumption, which includes 'green water' resources | China Statistical Yearbook, 2011–2020 [1]; Water Resources Bulletin of China, 2011–2020 [34] |
| | The number of employees in agriculture | China Statistical Yearbook, 2011–2020 [1] |
| | Cultivation area | China Statistical Yearbook, 2011–2020 [1] |
| | The total power of agricultural machinery | China Statistical Yearbook, 2011–2020 [1] |
| | Fertiliser consumption | China Statistical Yearbook, 2011–2020 [1] |
| Expected output indicator | Gross agricultural output value | China Statistical Yearbook, 2011–2020 [1] |
| Unexpected output indicators | Agricultural chemical oxygen demand emission | China Environmental Statistics Yearbook, 2011–2020 [35] |
| | Ammonia nitrogen emission | China Environmental Statistics Yearbook, 2011–2020 [35] |

### 2.4. DEA Model Input Series and Research Process

Agricultural water consumption (including green water resources), labour force, sowing area, agricultural machinery and fertiliser consumption were taken as input variables, and the total agricultural output value and total wastewater discharge (including total chemical oxygen demand (COD) discharge and total ammonia nitrogen discharge) were taken as output variables. The superefficiency DEA model was used to measure AWUE in five provinces in Northwest China (Table 2). On this basis, a modified gravity model was established to estimate the linkage strength of AWUE between any two provinces of Northwest China. The SNA method was adopted to analyse the spatial structure and network characteristics of AWUE in this region with the help of parameter indicators such as degree centrality, betweenness centrality, network density, core-edge structure and condensed subgroups. The specific research process is as follows:

(1) Considering the economic and geographical distance factors of the capital cities of five provinces in Northwest China, the modified gravity model is used to determine the spatial correlation between five provinces in Northwest China.

(2) To describes whether nodes are central in the network through degree centrality, betweenness centrality and closeness centrality.

(3) Using the network density and cohesive subgroup to reveal which provinces in Northwest China have closer connections and the impact of this agglomeration on the linkage network of AWUE in the whole region.

**Table 2.** DEA model input series.

| Region | Agricultural Water Consumption | Labour Force | Sowing Area | Agricultural Machinery | Fertiliser Consumption | Total Agricultural Output | Total Chemical Oxygen Demand Discharge | Total Ammonia Nitrogen Discharge |
|---|---|---|---|---|---|---|---|---|
| | $10^8$ m$^3$ | $10^4$ People | $10^3$ hm$^2$ | $10^4$ kw | $10^4$ t | $10^8$ Yuan | t | t |
| Shaanxi | 327.19 | 2 | 4195.96 | 2365.09 | 225.01 | 2937.64 | 381,978.70 | 12,154.40 |
| | 21.14 | 1 | 81.57 | 157.85 | 15.14 | 586.34 | 99,362.08 | 7694.95 |
| Gansu | 210.35 | 4 | 4032.01 | 2255.36 | 89.19 | 1639.27 | 379,499.60 | 16,919.70 |
| | 7.62 | 1 | 192.87 | 240.56 | 6.76 | 259.97 | 147,281.34 | 2771.03 |
| Qinghai | 38.79 | 1 | 556.86 | 453.98 | 8.47 | 352.21 | 486,900.40 | 19,535.90 |
| | 1.95 | 0 | 6.23 | 25.14 | 1.52 | 84.08 | 11,236.06 | 423.46 |
| Ningxia | 98.13 | 1 | 1218.37 | 708.64 | 39.42 | 497.30 | 410,884.50 | 18,760.30 |
| | 4.12 | 1 | 55.26 | 99.48 | 1.07 | 103.78 | 39,851.10 | 953.77 |
| Xinjiang | 623.71 | 39 | 5686.80 | 2440.38 | 232.66 | 3041.87 | 257,789.00 | 12,920.10 |
| | 32.22 | 23 | 456.35 | 368.41 | 28.14 | 736.61 | 176,836.49 | 5943.59 |

Note: The black number represents the average value from 2011 to 2020, and the grey number represents the standard deviation.

## 3. Results

### 3.1. AWUE Time Evolution Characteristics in Northwest China

Based on the superefficiency DEA model, the amount of "green water" resources is added to the input index of agricultural water usage, and the index of wastewater pollutants (total chemical oxygen demand discharge and total ammonia nitrogen discharge) is added to the undesirable output index to measure the changing trend of agricultural AWUE over time in the five provinces and regions in Northwest China (Figure 2). On

the whole, the agricultural AWUE in Northwest China shows an upwards trend. In 2011, the average value of agricultural AWUE was 0.4989. Although the utilisation of "green water" is considered, 50% of the water still does not play a role. The average AWUE in 2020 increased to 0.9582, nearly twice that in 2011.

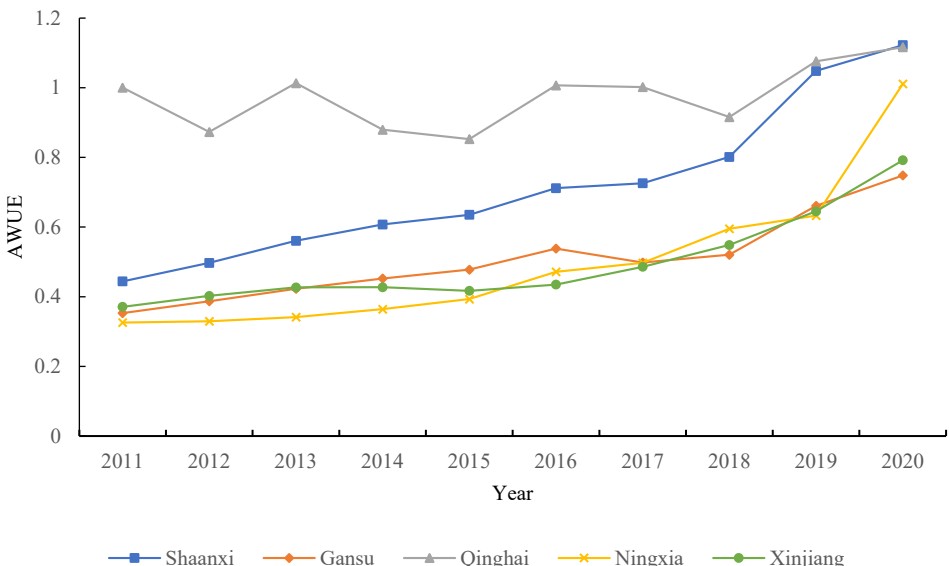

**Figure 2.** Temporal variation trend of AWUE in Northwest China.

From the changes in AWUE in Northwest China, the average value of agricultural AWUE in five provinces and regions from 2011 to 2020 is as follows: Qinghai (0.9735) > Shaanxi (0.7154) > Gansu (0.5059) > Ningxia (0.4963) > Xinjiang (0.4951). The natural precipitation in Qinghai Province is abundant, and the sown area of the main crops is small (accounting for 3.58% of the total sown area in Northwest China). Green water and blue water for irrigation can meet the water demand of agricultural grain, with high efficiency values and little overall variation. The annual precipitation in Shaanxi Province is 696.3 mm, approximately 1.9 times the average precipitation in Northwest China, so the value of AWUE is relatively high. The AWUE value of Ningxia increased from 0.6328 to 1.0112 from 2019 to 2020, which was related to a decrease in agricultural water use (8.3% of green water), an increase in expected output (gross agricultural product) and a decrease in undesirable output (wastewater pollutant index) in 2020. There is a lack of water resources in Gansu and Xinjiang due to the influence of climate, agricultural irrigation methods and agricultural technologies, and although the efficiency values increased to 0.7486 and 0.7920 in 2020, respectively, there is still room for improvement.

### 3.2. The Connection Strength of AWUE in Northwest China

In this study, the modified gravity model is used to measure the linkage strength between the AWUE of any two provinces (Figure 3). Overall, the linkage strength of AWUE in Northwest China showed a trend of rapid growth from 2011 to 2020, with a significant increase in 2020. The highest linkage intensity of AWUE between provinces in 2011, 2015, and 2020 was Qinghai-Gansu (2011, 2015) and Shaanxi-Gansu (2020). The results showed that Gansu and Qinghai were the important nodes of the AWUE linkage in 2011, while Ningxia and Xinjiang were marginalised, showing an isolated island phenomenon. In 2015, Shaanxi, Gansu and Qinghai were the important linking nodes of AWUE, and the isolated island status of Ningxia and Xinjiang gradually broke down. In 2020, Shaanxi and Gansu accounted for the largest proportion of connections, while Ningxia and Xinjiang showed a trend of rapid growth.

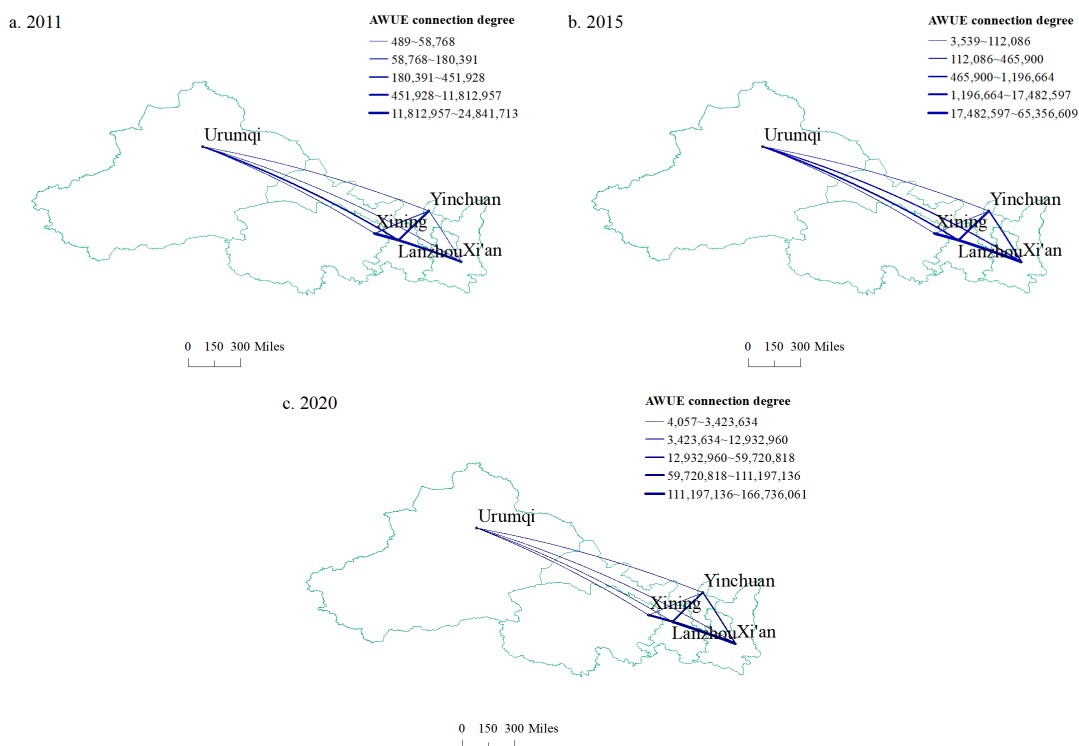

**Figure 3.** Correlation strength of AWUE. Legend: The thickness of the blue line in the figure indicates the strength of the connection.

### 3.3. The Overall Network Characteristics of AWUE in Northwest China

As shown in Figure 4, the AWUE linkage network density of the five provinces in Northwest China in 2011, 2015 and 2020 was 0.45, 0.65 and 0.8, respectively, indicating that the loosely connected network was becoming closely connected, and the linkage degree of AWUE continuously improved. The reason may be related to the impact of the provinces in the region. From Figure 3, it can be seen that the intensity of AWUE linkages in the northwest region shows a rapid growth trend from 2011 to 2020. The greater the interaction between the provinces, the higher the linkage tightness, and the greater the density of the AWUE network throughout the region. The core-edge model was used to analyse the locations of different provinces in the AWUE network in Northwest China. Shaanxi, Gansu and Qinghai were taken as the core provinces, and Ningxia and Xinjiang were taken as the edge provinces. The connection density between the core and edge members was 0.333, indicating that the network structure has an obvious hierarchical relationship. Further cohesive subgroup analysis showed that there was a strong correlation between AWUE in Shaanxi-Ningxia and Gansu-Qinghai (Figure 5).

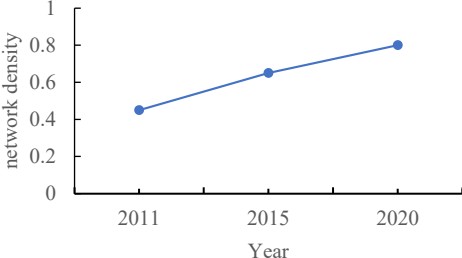

**Figure 4.** Network density.

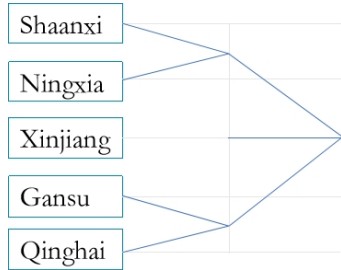

**Figure 5.** Cohesive subset.

*3.4. Individual Network Characteristics of AWUE in Northwest China*

Figure 6 shows that the relative degree centrality of the five provinces in Northwest China in 2011 was 45%, and the low-value areas were mainly distributed in Ningxia and Xinjiang with the lowest value of 25%, while the high-value areas were concentrated in Gansu with the highest value of 100%. This shows that the spatial difference in AWUE in Northwest China is significant, and the phenomenon of polarisation is serious. In 2015, the relative degree centrality was 65%, the low-value area was in Xinjiang with the lowest value of 37.5%, and the high-value area was in Shaanxi and Gansu with the highest value of 100%. This shows that the centrality of the AWUE network is enhanced, and the polarisation trend is alleviated. In 2020, the relative degree centrality was 80%, the low-value area was in Xinjiang, the lowest value was 50%, the high-value area was in Shaanxi and Gansu, and the highest value was 100%. This shows that the closeness of the AWUE network has been greatly improved, the radiation effect and driving ability of the core provinces have been significantly enhanced, and the overall network tends to develop in a balanced way.

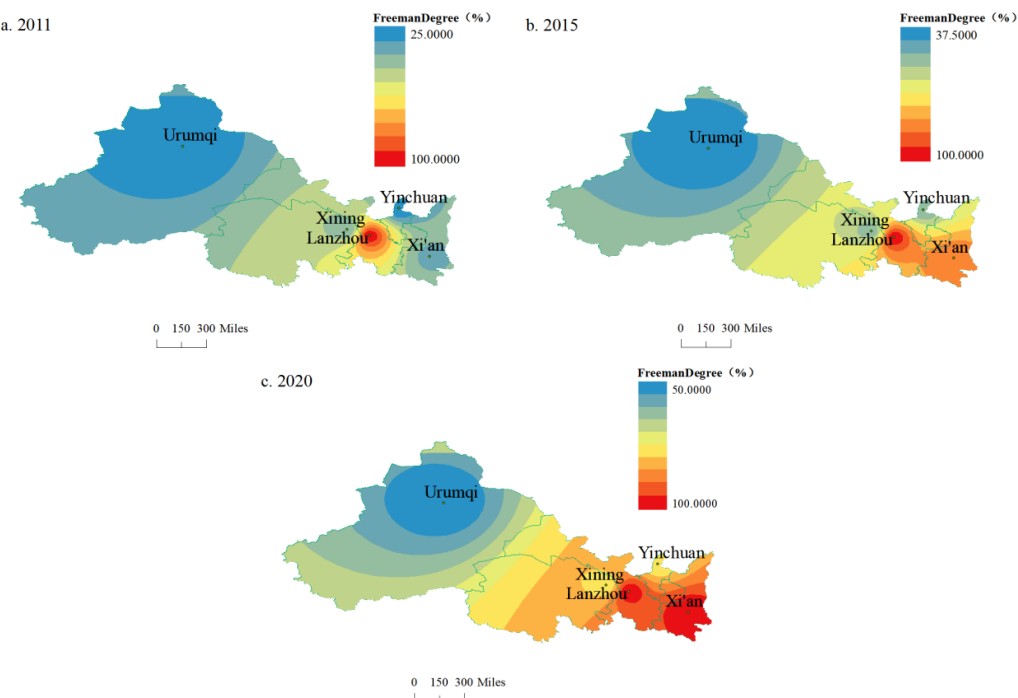

**Figure 6.** Degree centrality.

Table 3 shows that the point in degree and point out degree in 2011, 2015 and 2020 showed an overall upwards trend, indicating that the outflow and inflow among the five provinces in Northwest China gradually strengthened, and the AWUE connection between them became closer. Specifically, the point out degree of Shaanxi (2015) and Qinghai (2011) was higher than the point in degree, indicating that as the dominant node in the network, the node outputs more information to other nodes, and these nodes had stronger trading

and radiation ability in the AWUE link network, which was more efficient and convenient for transmitting and exporting related resource elements, as it was in the centre of the network. The point in degree of Shaanxi (2011) and Xinjiang (2015) was higher than the point out degree, which means that these nodes were more greatly affected by the power provinces than by themselves.

**Table 3.** Network connection direction.

| Year | Degree | Shaanxi | Gansu | Qinghai | Ningxia | Xinjiang | Average |
|------|--------|---------|-------|---------|---------|----------|---------|
| 2011 | InDegree | 2 | 4 | 1 | 1 | 1 | 1.8 |
|      | OutDegree | 1 | 4 | 2 | 1 | 1 | 1.8 |
| 2015 | InDegree | 3 | 4 | 2 | 2 | 2 | 2.6 |
|      | OutDegree | 4 | 4 | 2 | 2 | 1 | 2.6 |
| 2020 | InDegree | 4 | 4 | 3 | 3 | 2 | 3.2 |
|      | OutDegree | 4 | 4 | 3 | 3 | 2 | 3.2 |

As shown in Figure 7, the relative betweenness centrality of Gansu in 2011 was 91.67%, which was much higher than that of other provinces, indicating that Gansu was in a monopolistic position in the AWUE linkage network in Northwest China. Regional AWUE was highly dependent on its intermediary role, and the AWUE linkage network was unbalanced and lacking in stability. In 2015 and 2020, the relative intermediate centrality of Shaanxi and Gansu was large, and the intermediary effect was strong, forming an agglomeration effect.

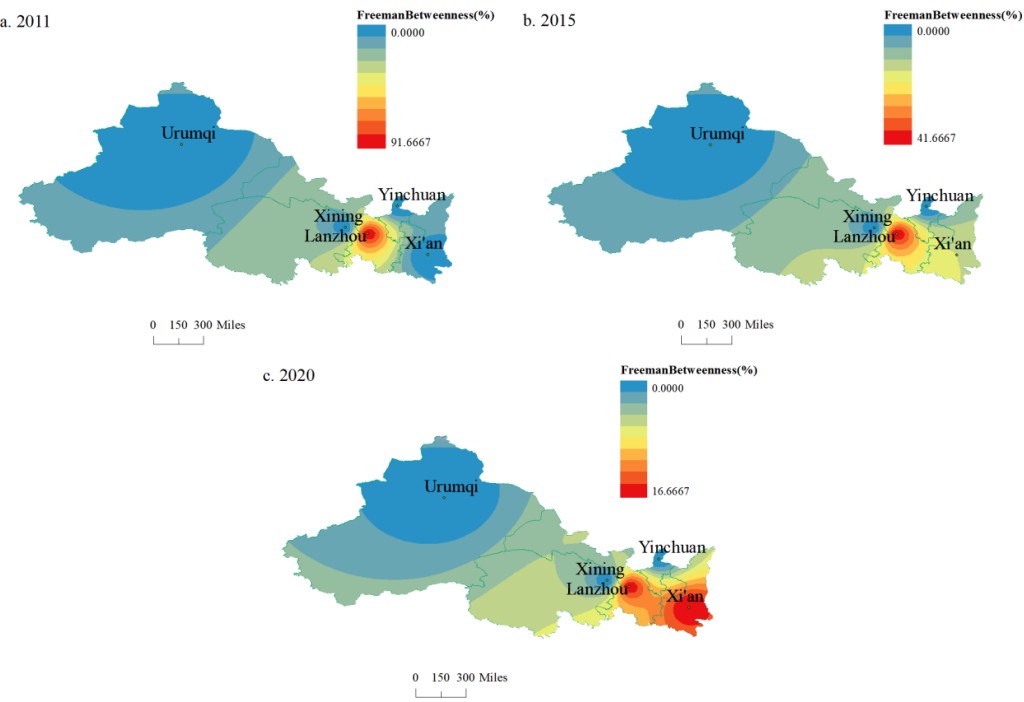

**Figure 7.** Betweenness centrality.

## 4. Discussion

### 4.1. The Overall AWUE in Northwest China Is on the Rise

AWUE in Northwest China demonstrated an overall upwards trend from 2011 to 2020. The AWUE of Shaanxi, Qinghai and Ningxia effectively reached the frontier of DEA in 2020. The efficiency loss caused by excessive agricultural input, insufficient expected yield and improper resource allocation was basically overcome in this region in 2020. Although the AWUE values of Gansu and Xinjiang indicated that they were ineffective provinces, according to DEA (efficiency values below 0.8), there was an increasing trend from 2011 to 2020 and great potential to improve AWUE. Compared with the research results of

Zhang [36], Liu [37] and Sun [38], the AWUE value in Northwest China obtained in this study was slightly higher, which may be because most of the previous studies focused on only 2016 or earlier, while the research period of this study was nearly 10 years (2011–2020). As seen from the results (Figure 2), AWUE in the five provinces in Northwest China was in a rapid development stage after 2016, so there were some differences from the previous research results. In this study, the variation in AWUE in provinces and regions in Northwest China was basically the same as that of Zhang et al. [36–38]. The AWUE value in Qinghai was the highest, the AWUE value in Xinjiang was the lowest, and the other three provinces had values in the middle. Overall, the results of this study are consistent with those of other studies.

### 4.2. The Connection Tightness of AWUE in Northwest China Is Enhanced

The AWUE network density and degree centrality of the five provinces in Northwest China increased from 0.45 to 0.8 from 2011 to 2020, indicating that the closeness of water use efficiency links gradually increased. The increase in network density is related to the influence of each province in the region. The greater the mutual influence between provinces, the higher the connection density, and the greater the network density of water use efficiency in the whole region. Degree centrality can represent the synthesis of a province's direct connection with other provinces. The larger the value is, the greater the degree of interaction between provinces. The connection strength is the calculation base of network density and degree centrality. Figures 2 and 8 show that the increase in water use efficiency, total population, actual regional GDP and per capita GDP in the capital cities of the five provinces in Northwest China led to the rapid growth of water use efficiency in Northwest China from 2011 to 2020 (Figure 3). The growth of the connection strength eventually increased the network density and the degree centrality of the northwest region, that is, the connection tightness of the overall network.

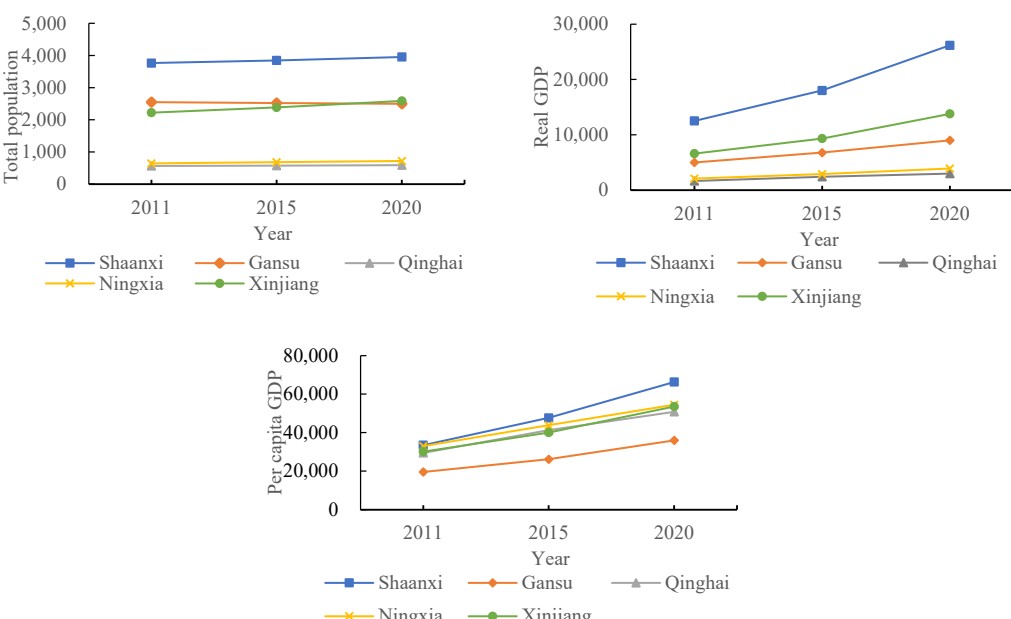

**Figure 8.** Population, real GDP and per capita GDP in Northwest China.

### 4.3. Discussion on the Path of Improving AWUE in Northwest China

In the process of water resource planning and policy implementation, we should not only grasp the linear causal relationship of water use efficiency but also grasp the spatial correlation characteristics of regional overall water use efficiency from the interactive relationship of the water use efficiency level among regions. Most of the existing studies [39–41] focus on the national and provincial scales. Although the national scale

can control the overall AWUE in China, due to the vast territory of China, the supply and demand of water resources in different regions vary greatly. In particular, regional difference in utilisation efficiency between Northwest China, a typical arid area, and Southeast China, where water resources are abundant, are difficult to reflect. Studies at the regional scale are usually based on the humid and subhumid regions in Central and Eastern China. Due to vast differences in objective conditions within Northwest China and Central and Eastern China, evaluating the efficiency of Northwest China and Central and Eastern China in accordance with the same optimal frontier is unreasonable. Under the process of marketisation and integration, the utilisation of water resources in Northwest China is showing an increasingly multi-threaded and complicated development trend. Based on the inter-provincial linkage effect of water resources and the perspective of spatial network, this study provides a new idea for improving the overall AWUE in Northwest China. The spatial correlation network analysis showed that the correlation intensity and tightness of AWUE in Northwest China were enhanced. It also presents a clear 'core-edge' structural feature and is relatively stable. Among them, Shaanxi and Gansu have high-degree centrality and intermediary centrality and are in the position of central actors in the network. They are the main forces that promote the optimisation of overall agricultural water use efficiency and the evolution of the spatial network in Northwest China. In fact, the development of water resources in Northwest China is relatively high, the utilisation mode of water resources is extensive, the utilisation efficiency is low, and the potential of new water sources in Gansu and Shaanxi provinces is very small. In addition to Shaanxi, the average water consumption per acre of farmland in Gansu and Qinghai is 33% higher than that in the whole country, that in Xinjiang is 65% higher than that in the whole country, and that in Ningxia is 2.6 times higher than that in the whole country, so the water saving potential of farmland is enormous [42]. Combined with the results of cohesive subgroup analysis, to improve the AWUE of the whole network in Northwest China, it is necessary to focus on promoting regional cooperation on AWUE between Shaanxi–Ningxia and Gansu–Qinghai. Based on this, the study proposes policy recommendations for gradually realising the water use efficiency improvement path of the central province (Shaanxi; Gansu)–cooperation circle (Shaanxi–Ningxia; Gansu–Qinghai)–the whole region.

## 5. Conclusions

Based on panel data on AWUE in Northwest China from 2011 to 2020, this study uses a superefficiency DEA model that accounts for the 'green water' index and undesired output index to explore the AWUE and its spatial network structure characteristics in Northwest China under changes in climate factors such as precipitation. The study points out that it is necessary to pay attention to coordination and cooperation in water resource utilisation among the five provinces in Northwest China, give full play to the linkage role of core provinces, drive the absorption capacity of marginal provinces to water resources, economy, technology and other elements, strengthen the cooperative relationship with neighbouring provinces, improve the overall AWUE level in Northwest China, narrow the differences among provinces, and achieve spatially balanced development. There are two shortcomings in the research. First, the calculation of the connection intensity of AWUE in Northwest China only selects the water use efficiency value, total population, real GDP and per capita GDP of the provincial capital cities, and the contribution of other cities in the province to the connection intensity still needs to be discussed in depth. Second, the analysis of AWUE in Northwest China has not been linked to the national AWUE in China. The above deficiencies need to be further improved in future research.

**Author Contributions:** Conceptualization, Y.G. and P.L.; methodology, Y.G.; software, Y.Z.; validation, K.L.; writing—original draft preparation, Y.G.; writing—review and editing, X.Q.; funding acquisition, Y.G. and P.L. All authors have read and agreed to the published version of the manuscript.

**Funding:** This research was funded by the National Key R&D Program of China (no. 2022YFD1900502), National Natural Science Foundation of China (no. 52309073), Henan Province Key R&D and Promo-

tion Special Project (no. 222102110385), and Agricultural Science and Technology Innovation Program of Chinese Academy of Agricultural Sciences (no. CAAS-ASTIP).

**Data Availability Statement:** All data generated or analysed during this study are included in this published article.

**Conflicts of Interest:** The authors declare no conflict of interest. For my coauthors, I declare that the described work is original research that has not been published previously and is not under consideration for publication elsewhere, in whole or in part.

## Abbreviations

AWUE    agricultural water use efficiency
DEA    data envelopment analysis
SNA    social network analysis

## Appendix A. Degree Centrality

Degree centrality can describe the cohesion degree of a province in the whole AWUE network, which is the synthesis of the direct connection between a province and other provinces. The degree of a node includes the point out degree and the point in degree. The out degree is the expansion attribute, which refers to the number of edges emitted from the node, indicating the influence of the province on the WUE of other provinces. The larger the value, the greater the influence and the radiation range of the province. The in degree is the receiving attribute, which represents the number of edges pointing to the node from other nodes and is the degree to which the province is affected by the AWUE of other provinces. The calculation formula is as follows:

$$C_{RD}(i) = \frac{C_{AD}(i)}{n-1}$$

where $C_{RD}(i)$ refers to the relative degree centrality of node $i$ and $C_{AD}(i)$ is the absolute degree centrality of node $i$. Both are used to express the connection state of each point in the network. The former can better show the situation of each point in the network, while the latter is used to calculate the number of other points connected with point i in the network.

## Appendix B. Betweenness Centrality

Betweenness centrality represents the number of times that a node acts as an intermediary to help any other two nodes connect with each other in the shortest path. The more times a node acts as an intermediary in the network, the greater the betweenness centrality. The calculation formula is as follows:

$$C_{RB}(i) = \frac{2C_{AB}(i)}{(n-1)(n-2)} = \frac{2\sum_j^n \sum_k^n b_{jk}(i)}{(n^2 - 3n + 2)} = \frac{2\sum_j^n \sum_k^n g_{jk}(i)/g_{jk}}{(n^2 - 3n + 2)}$$

where $C_{RB}(i)$ is the relative betweenness centrality of point $i$; $C_{AB}(i)$ is the absolute betweenness centrality; $b_{jk}(i)$ represents the ability of point $i$ to control the interaction between points $j$ and $k$; $g_{jk}$ represents the number of shortcuts between points $j$ and $k$; and $g_{jk}(i)$ represents the number of shortcuts through point $i$ between points $j$ and $k$.

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
