# Peer review of "Analysis of the Spatial Correlation Network Structure of Agricultural Water Use Efficiency in Northwest China"

_agronomy, doi:10.3390/agronomy13102509_

Round 1

Reviewer 1 Report

The manuscript titled ‘Analysis of the spatial correlation network structure of agri-2 cultural water use efficiency in Northwest China’ is considered an extremely relevant and scientific topic in an adequate way. Notwithstanding, some recommendations may be added to increase the value of the manuscript.

The abstract should be revised partly indicating that the empirical findings may be demonstrated in more detail. Moreover, the theoretical chapters of underpinnings, method, and the first part of results are regarded as acceptable, however, from 3.2.3. the analysis slightly is superficial. More precisely, in these sections, the manuscript must describe why network density was increasing between 2011 and 2020. The manuscript states that it happened but the reader does not understand why it emerged. Originally, the manuscript aims to explain this process, but we have so far not received the answer in the paper.

To sum up, the manuscript covers a novel scientific topic, but some points need to be modified in order to be published in the Agronomy Journal. Provided the authors revised and corrected above mentioned sections, this manuscript can be published.

The English terminal terms are applied sophistically by authors. 

Author Response

Dear reviewer, thank you very much for your comments, which are very valuable to improve the quality of this paper. We have made changes based on your comments and have improved the following: Reviewer #1:

1. The summary should be partially revised to show that empirical findings can be presented in greater detail. Reaction: We have revised the content of the abstract. For revisions, please refer to line 28-1 on page 6.

2. The analysis from 3.2.3 is slightly superficial. More precisely, in these sections, the manuscript must describe why network density increased between 2011 and 2020. The manuscript indicates that it happened, but the reader doesn't understand why it happened. Initially, the manuscript was intended to explain this process, but so far we haven't got the answer in the paper. Reaction: The contents of section 3.2.3 of the original manuscript have been revised and supplemented, as shown in sections 3.3 and 4.2 of the revised manuscript. For the revision, see page 266, lines 270-8, and page 342, lines 360-11.

Reviewer 2 Report

Review report

Title: Analysis of the spatial correlation network structure of agricultural water use efficiency in Northwest China

Abstract

-The abstract should be concise and summary of the major findings.

I recommend the authors rephrase the abstract.

The results stated in the abstract can be shortened and only the major results must be presented.

Further, it should be presented at least in two paragraphs rather than congesting the whole text as one paragraph.

Introduction

Paragraph 1

Line 59 (the numeric figures are not referenced) It is better to cite the source of the data.

Paragraphs 1 and 2 can be combined and concise.

Methods

In the introduction part, the authors mentioned they used a data panel from 2011-2020, whereas in 2.2.1. section, they used from 2010-2020?

Section 3.

Better to only write Results rather than 3. Results and analysis

This figure does not mention the x and y axis, well X axis seems year but what about the Y axis?

Fig. 2. Temporal variation trend of WUE in Northwest China

In Fig 3, the legends are not clear or visible.

Discussion

Line 383

What are those studies (it is better to cite those previous studies)?

In the conclusion part

The conclusion did not include the implication of the research. Further the limitation of the research work?

Dear Authors,

The English grammar and structure of the manuscript is fine, there is minor editing required. Some of the sentences are too long which decrease readability.

Author Response

Dear reviewers, thank you very much for your comments, which have been highly valuable for improving the quality of this paper. We have made changes based on your comments, and we have improved the following:

Reviewer #2:

  1. The abstract should be concise and summary of the major findings. I recommend the authors rephrase the abstract. The results stated in the abstract can be shortened and only the major results must be presented.

Responses:

We have modified the content of the abstract. The results stated in the abstract are simplified, and only the main results are presented.

Please refer to lines 6-28 on page 1 for revisions.

  1. Paragraph 1. Line 59 (the numeric figures are not referenced) It is better to cite the source of the data. Paragraphs 1 and 2 can be combined and concise.

Responses:

The numbers in the first paragraph come from the 'China Statistical Yearbook' and 'China Water Conservancy Statistical Yearbook', which have been added to the article. The content of the first paragraph has been simplified. The second paragraph has been deleted, and the background of the study has been supplemented.

Please refer to lines 36-38 on page 1 and lines 48-62 on page 2 for revisions.

  1. In the introduction part, the authors mentioned they used a data panel from 2011-2020, whereas in 2.2.1. section, they used from 2010-2020? Section 3. Better to only write Results rather than 3. Results and analysis. Fig. 2. does not mention the x and y axis, well X axis seems year but what about the Y axis? In Fig 3, the legends are not clear or visible.

Responses:

Section 2.2.1 has made a mistake and has revised 2010-2020 to 2011-2020. 3. Results and analysis has been changed to 3. Results, and the structure and content of this part have been modified. The axis titles of Figure 2 and Figure 4 have been added, and the numbers and legend numbers in Figure 3, Figure 6 and Figure 7 have been increased.

Please refer to line 155 on page 4, Section 3 and Figures 2 to 7 for revisions.

  1. Line 383. What are those studies (it is better to cite those previous studies)?

Responses:

References have been added to 'those studies'.

Please refer to line 339 on page 10 for revisions.

  1. The conclusion did not include the implication of the research. Further the limitation of the research work?

Responses:

The conclusion section has been modified. It increases the meaning of the research and the limitations of the research.

Please refer to lines 385-403 on page 12 for revisions.

Reviewer 3 Report

GENERAL COMMENTS 

The manuscript entitled “Analysis of the spatial correlation network structure of agricultural water use efficiency in Northwest China " investigates how a Data Envelope Model (DEA) is appropriate to measure the Water Use Efficiency (WUE) in five provinces in Northwest China. I recognize the relevance of this research to address a knowledge gap in the field and the potential applications of the outputs of this study on the benefit of the region. I do observe that there is a lot of work developed; however, the manuscript is poorly written, with fragmented ideas that difficult the reading and proper appreciation of this research. 

In addition, the structure of this manuscript is ill-suited; for example, the abstract is too long and described methodology and results in too much detail without extracting the essence of the research carried out. On the other hand, the “Introduction” does not set the context of this research, instead it is comprised of a brief literature review plus the description of the area of study and the presentation of importance of West China in the national agricultural sector.

Finally, the resolution of figures should be increased, and the sections of “discussion” and “conclusions” should be strengthened. 

no

Author Response

Dear reviewers, thank you very much for your comments, which have been highly valuable for improving the quality of this paper. We have made changes based on your comments, and we have improved the following:

Reviewer #3:

  1. The structure of this manuscript is ill-suited; for example, the abstract is too long and described methodology and results in too much detail without extracting the essence of the research carried out.

Responses:

We have modified the structure of this manuscript. The content of the abstract has been revised. The methods and conclusions in the abstract have been reduced, and only the main results are presented.

Please refer to lines 6-28 on page 1 for Abstract revisions.

  1. The “Introduction” does not set the context of this research, instead it is comprised of a brief literature review plus the description of the area of study and the presentation of importance of West China in the national agricultural sector.

Responses:

It has revised the introduction, added the background of the research, deleted the introduction of western China, and added the research of other countries.

Please refer to lines 48-62 on page 2 and lines 109-118 on page 3 for revisions.

  1. The resolution of figures should be increased.

Responses:

The numbers and legend numbers in Figure 3, Figure 6 and Figure 7 have been increased.

Please refer to Figures 3 to 7 for revisions.

  1. The sections of “discussion” and “conclusions” should be strengthened.

Responses:

The conclusion and discussion sections have been modified.

Please refer to lines 323-403 on pages 10-12 for revisions.

Reviewer 4 Report

The article aims to analyze the spatial correlation network structure of agricultural WUE  in Northwest China. The idea is interesting, still, some points should be modified.

First, the Introduction refers only to studies done in China. I imagine that this topic was studied in other countries as well. It would be interesting to know where and with what results.

The citations are made in text with superscript letters, which is not in concordance with the template. The references are not in the correct template, as well.

The introduction must emphasize the study's novelty. It does not result clearly from the presentation.

Fig. 1 must be Figure 1, according to the template. The same for other figures. The note should be placed after the figure's caption.

The variables must be in Italics in the manuscript.

Lines 77-78, please be consistent with the notations in the equation - r1, r2 must be r1, r2 etc.  X X ∈ (Rm) -it should be a mistake.

Please check all the equations and the corresponding significance of the variables in the text.

The presentation of the methods is not well done. It must be polished to be more coherent.

What does it mean 'upwards trend in bands'? 

Lines 274-276 should be moved at the presentation o the region.

Figure 3 must be redone to eliminate the big blank spaces between the figures. Please explain the significance of the blue lines in the figure. Same remark for Figures 4,5. Please also increase the letters since the writing in the figures can be difficult to distinguish.

In Figures 2, 6,7 the letters are excessively high.

'the point out degree of Shaanxi (2015) 326 and Qinghai (2011) was higher than the point in degree' - If I understood well the significances of the variables, the assertion is not correct, according to Table 2.

In Conclusion, you should not repeat what is already in the Results and Discussion but summarize the importance of the study.

Overall, the article is carelessly written and does not point out the novelty of the research. It must be completely rewritten before being reevaluated for publication.

Moderate English corrections are required.

Author Response

Dear reviewers, thank you very much for your comments, which have been highly valuable for improving the quality of this paper. We have made changes based on your comments, and we have improved the following:

Reviewer #4:

  1. The Introduction refers only to studies done in China. I imagine that this topic was studied in other countries as well. It would be interesting to know where and with what results. The citations are made in text with superscript letters, which is not in concordance with the template. The references are not in the correct template, as well. The introduction must emphasize the study's novelty. It does not result clearly from the presentation.

Responses:

We have added research from scholars from other countries to the introduction. The format of citations and references was modified. The content of the research background has been added to the introduction, that is, the consideration of 'green water' resources to emphasize the novelty of this study.

Please refer to lines 109-118 on page 3, lines 48-62 on page 2, and References for revisions.

  1. Fig. 1 must be Figure 1, according to the template. The same for other figures. The note should be placed after the figure's caption.

Responses:

Fig. 1 has been modified to Figure 1, and other figures have also been modified. The figure legends have been placed after the figure titles.

Please refer to Figures 2 to 8 for revisions.

  1. The variables must be in Italics in the manuscript. Lines 77-78, please be consistent with the notations in the equation - r1, r2 must be r1, r2 etc.  X X ∈ (Rm) -it should be a mistake. Please check all the equations and the corresponding significance of the variables in the text.

Responses:

The variables in the manuscript have been modified to italics. 'X X' belongs to a mistake and has been modified. All equations and corresponding meanings of the variables have been checked and verified.

Please refer to lines 160-165, 167-168, and 178-184 on page 5 and lines 512-516 and 522-525 on page 14-15 for revisions.

  1. The presentation of the methods is not well done. It must be polished to be more coherent.

Responses:

Rewrite the method part.

Please refer to Section 2.2 for revisions.

  1. What does it mean 'upwards trend in bands'?

Responses:

The 'upwards trend in bands' has been revised to an upwards trend.

Please refer to line 225 on page 6 for revisions.

  1. Lines 274-276 should be moved at the presentation o the region.

Responses:

The text content of lines 274-276 in the original manuscript has been moved to Figure 2.

Please refer to lines 228-243 on pages 6-7 for revisions.

  1. Figure 3 must be redone to eliminate the big blank spaces between the figures. Please explain the significance of the blue lines in the figure. Same remark for Figures 4,5. Please also increase the letters since the writing in the figures can be difficult to distinguish. In Figures 2, 6,7 the letters are excessively high.

Responses:

The number and legend size of Figure 3 have been increased, and the meaning of the blue line has been annotated. The numbers and legends of Figure 6 and Figure 7 (Figure 4 and Figure 5 in the original draft) have been increased, and letters have been added to the picture to distinguish them. The text size in Figure 2, Figure 4 and Figure 5 (Figure 6 and Figure 7 in the original manuscript) has been modified.

Please refer to Figures 2 to 7 for revisions.

  1. 'the point out degree of Shaanxi (2015) 326 and Qinghai (2011) was higher than the point in degree' - If I understood well the significances of the variables, the assertion is not correct, according to Table 2.

Responses:

In Table 2, the out-degree value of Shaanxi in 2015 was 4 higher than the in-degree value of 3, and the out-degree value of Qinghai in 2011 was 2 higher than the in-degree value of 1.

Please refer to line 304 on page 9 for revisions.

  1. You should not repeat what is already in the Results and Discussion but summarize the importance of the study.

Responses:

The conclusions and discussions have been revised, and the importance of the research has been summarized in the conclusions.

Please refer to lines 323-403 on pages 10-12 for revisions.

Round 2

Reviewer 3 Report

GENERAL COMMENTS 

These comments applied to the second version of the manuscript entitled “Analysis of the spatial correlation network structure of agricultural water use efficiency in Northwest China ", authors have addressed all observations made to the original document, thus the structure of this manuscript has greatly been improved, the abstract was shortened and rewritten to the point. The sections of “discussion” and “conclusions” were strengthened, the redaction of the whole manuscript was improved, and the resolution of figures were increased. 

Just one comment, the “Introduction” could be bettered, the first paragraph (rows 33 to 47) starts with a description of the problem in China, this is the shortage of water resources and the agricultural water use efficiency (AWUE) in the northwest region in China. It is recommendable to move this paragraph to line 119, before the final paragraph that closes the introduction section.

no

Author Response

Dear reviewer, thank you very much for your comments. We have made changes based on your comments:
The first paragraph (lines 33 to 47) of the original manuscript has been moved to lines 104-118, which before the final paragraph that closes the introduction section.

Reviewer 4 Report

Dear Authors,

I read your manuscript with interest.

Some remarks are necessary.

Please introduce the chartflow of the study.

Please include the statistical analysis of the input series

Typos are still present at pages 4 and 5. For example, ''1, , m'', should be ''1,...,m'', line 167 you should have Pe, not Pe.

The formula after line 166 should be in italics. Please revise the entire manuscript.

References are not in the correct format.

 Moderate editing of English language required.

Author Response

Dear reviewer, thank you very much for your comments, which have been highly valuable for improving the quality of this paper. We have made changes based on your comments, and we have improved the following:

1. Please introduce the chart flow of the study. Please include the statistical analysis of the input series.

Responses:

We have added Section 2.4 to illustrate DEA model input series and research process.

Please refer to lines 217-239 on page 6-7 for revisions.

2. Typos are still present at pages 4 and 5. For example, ''1, , m'', should be ''1,...,m'', line 167 you should have Pe, not Pe.

Responses:

We have modified the typos.

Please refer to lines 159 and 167 on page 4-5 for revisions.

3. The formula after line 166 should be in italics. Please revise the entire manuscript. References are not in the correct format.

Responses:

We checked the formula in the manuscript, which is italic. And we abbreviated the journal name of the reference.